# Ultrafast one-minute electronic detection of SARS-CoV-2 infection by 3CL^pro enzymatic activity in untreated saliva samples

Ella Borberg[1], Eran Granot[1] & Fernando Patolsky [1,2] ✉

Since its onset in December 2019, severe acute respiratory syndrome coronavirus 2, SARS-CoV-2, has caused over 6.5 million deaths worldwide as of October 2022. Attempts to curb viral transmission rely heavily on reliable testing to detect infections since a large number of transmissions are carried through asymptomatic individuals. Many available detection methods fall short in terms of reliability or point-of-care applicability. Here, we report an electrochemical approach targeting a viral proteolytic enzyme, 3CLpro, as a marker of active infection. We detect proteolytic activity directly from untreated saliva within one minute of sample incubation using a reduction-oxidation pH indicator. Importantly, clinical tests of saliva samples from 50 subjects show accurate detection of SARS-CoV-2, with high sensitivity and specificity, validated by PCR testing. These, coupled with our platform's ultrafast detection, simplicity, low cost and point-of-care compatibility, make it a promising method for the real-world SARS-CoV-2 mass-screening.

COVID-19 (Coronavirus disease 2019), the disease caused by severe acute respiratory syndrome coronavirus 2 (SARS-CoV-2), was acknowledged by the World Health Organization (WHO) as a pandemic outbreak in March 2020[1], causing over 6.5 million deaths worldwide as of October 2022[2], with worldwide health and economic effects that are expected to persist for years to come[3]. To end the pandemic, attempts to lower transmission rates have been implemented. Unfortunately, SARS-CoV-2 transmission restriction by traditional countermeasures, based on isolating symptomatic individuals, is ineffective since a large percentage of infections is caused by asymptomatic carriers[4–6], thus, counteractions taken to curb the COVID-19 pandemic depend on strict strategies for high-quality testing procedures. These procedures mainly target specific viral molecules for identifying infected carriers[7].

SARS-CoV-2 is a coronavirus of the family *Coronaviridae*, and it is an enveloped positive-sense single-stranded ribonucleic acid (RNA) virus[1]. The four structural proteins are spike, envelope, membrane, and nucleocapsid. Spike protein mediates entry into host cells by binding to a cellular receptor, angiotensin-converting enzyme 2[8]. Then Spike protein is cleaved by cellular cathepsin L and the transmembrane protease serine 2[9]. Following the release of the viral genome into the host cytosol, replicase genes in the open reading frames of the RNA are translated into viral replicase polyproteins, which are cleaved into individual non-structural proteins via host and viral proteases such as 3-chymotrypsin-like protease (3CL^pro)[10,11]; forming the RNA polymerase[12]. Replicase components additionally cause a change in the endoplasmic reticulum forming double-layered vesicles, facilitating viral genomic replication and virion formation[11,13].

Currently, the primary detection methods include reverse transcription-polymerase chain reaction (RT-PCR), a well-established sensitive diagnostic method nearly reaching single-molecule sensitivity[14] and uses non-invasive sampling such as saliva or throat and nasal swabs. However, RT-PCR requires high-end equipment, unsuitable for point-of-care (POC) settings, and intricate sample processing by specialized lab personnel, thus representing a time-consuming approach. The required logistics delays screening results even further, typically given after several hours or even days after sampling, hindering the timeliness of reactive measures and thus allowing for larger transmission chains[15]. In addition, PCR is susceptible to foreign nucleic acid contamination non-specific amplification[16], and the presence of

[1]School of Chemistry, Faculty of Exact Sciences, Tel Aviv University, Tel Aviv 69978, Israel. [2]Department of Materials Science and Engineering, the Iby and Aladar Fleischman Faculty of Engineering, Tel Aviv University, Tel Aviv 69978, Israel. ✉e-mail: fernando@post.tau.ac.il

viral genomic material alone does not indicate active infection, possibly marking non-infectious individuals[17,18].

While immunoassays, such as enzyme-linked immunoassays (ELISA), are extremely sensitive approaches for protein detection and quantification, with protocols that are much less time-consuming than PCR, their mechanism is based on antigen–antibody interactions, making specific and high-affinity antibodies essential. These antibodies could be expensive, thus restricting the application of immunoassays as widespread POC measures. In order to address this issue, several experimental studies have implemented low-cost analogs of antibodies with promising results[16].

Multiple antibody tests have been developed to detect SARS-CoV-2, including lateral flow immunoassay (LFIA), chemiluminescence enzyme immunoassay (CLIA), and fluorescence enzyme-linked immunoassay (FIA). The majority of these assays use spike or nucleocapsid proteins of SARS-CoV-2[19,20] to detect immunoglobulin G (IgG) and/or immunoglobulin M (IgM) antibodies produced by the host immune system against the virus. The reported methods are relatively fast (several minutes), and many are compatible with POC approaches. However, the most applicable test for POC approaches, LFIA, reportedly has the lowest performance[21], and these quantitative and qualitative assays detect exposure to SARS-CoV-2 by antibody responses rather than active infection. This can aid in identifying factors that correlate with effective immunity to SARS-CoV-2[22], but it is less suitable for diagnosing infectious individuals[23].

Since 3CL$^{pro}$ is a non-structural protein, it is not exposed to the viral particle; therefore, it is not prone to linger in host fluids as viral envelope fragments. Moreover, since 3CL$^{pro}$ carries out a critical function in viral replication, its activity is essential for the viral life cycle; thus, its presence indicates active viral replication[11]. As a critical part of viral proliferation, meaning active infection, 3CL$^{pro}$ has been extensively studied in coronaviruses, past and current, as a target for treatment[24–26]. 3CL$^{pro}$ is a viral proteolytic enzyme that belongs to the cysteine protease class[27,28] and acts as a catalyst for peptide bond hydrolysis of viral polyproteins. SARS-CoV-2 proteins are expressed as two polypeptides that are polyprotein chains cleaved in specific sites[29]. 3CL$^{pro}$ cleaves at specific sites of amino acid sequences, usually in the LeuGln*Ser pattern. Ser could be replaced with either Ala or Gly (the cleaving site is marked with *)[26,30]. The 3CL$^{pro}$ catalytic site holds a catalytic dyad of Cys145-His41. The hydrolysis is catalyzed in a well-known nucleophilic reaction. First, Cys thiol is deprotonated by His residue, causing a nucleophilic attack of the substrates carbonyl carbon by the anionic sulfur, followed by the N-terminus of the substrate being protonated by the His residue of the catalytic site and detaching from the substrate. The C-terminus of the substrate forms a thioester intermediate with Cys residue, which is then hydrolyzed to produce a carboxylic acid and regenerate the catalytic site. The carboxylic acid product may cause an in vitro pH drop in a non-buffered medium[31,32].

Proteases have been recognized as essential biomarkers in many conditions, including cancer[33], Alzheimer's[34], AIDS[35], and inflammation[36]. Leading to targeting proteases as a target of drugs and as a diagnostic tool[37]. Protease detection assays could be grouped into affinity and activity assays. Since affinity assays detect protease regardless of activity, activity assays are more applicable for functional protease detection. Activity assays include colourimetric[38], mass spectrometry-based[39], and fluorescence resonance energy transfer assays[40]. These can achieve low detection limits (-p$_M$) but cannot be applied in multiplexed sensing platforms since only a few probes can generate different signals. More recently, nanomaterials such as noble metal nanoparticles[41], quantum dots[42], and graphene oxide[43] have been introduced in protease assays with impressive detection limits and more multiplexing capabilities. However, these are prone to limitations in the stability of the reporter molecules.

An additional group of assays, where the substrate is immobilized on the array's surface, includes electrochemical[44], surface-enhanced Raman scattering[45], and surface plasmon resonance assays[46]. These provide a platform for protease detection that could be easily multiplexed. Nonetheless, the sensitivity of these assays tends to be lower due to the substrate immobilization onto the detection surface, causing only proteases near surfaces to elicit a signal.

Over the past few decades, there has been an upsurge in the use of electrochemical biosensors for analysis of food quality[47,48], environmental monitoring[49,50], and clinical diagnostics[51,52]. Electrochemical biosensors display higher selectivity and sensitivity, faster response times, and require a lower amount of sample volumes when compared to traditional standard methods[53,54]. These attributes, combined with operational simplicity, cost-effectiveness, multiplexing capability, and possibility for miniaturization, lead to electrochemical biosensors being frequently designed for POC analysis[52,55].

With the development of micro-and nanofabrication, complex benchtop instruments, such as conventional potentiostats, have been miniaturized to allow POC measurements to be performed by untrained individuals[47]. Miniaturized electroanalytical devices, as small as a cellphone, have been commercially available as POC electrochemical biosensors for decades. The leading example is the glucometer, first introduced as a table-sized system by Clark et al. in 1962[56]. These widely used diagnostic devices are coupled to a pocket-sized amperometric transducer[52] and have been the leading diabetes monitoring devices in the market over the past decades, used both by physicians and patients[57]. In addition, electrochemical biosensors possess most of the ASSURED criteria set by the WHO to qualify as efficient diagnostic testing: Affordable, Sensitive, Specific, User-friendly, Rapid and robust, Equipment-free and Deliverable to end-users, making them prime candidates for SARS-CoV-2 detection[58].

In this work, we target 3CL$^{pro}$ as a specific biomarker of active viral replication. Both the presence and activity of 3CL$^{pro}$ in saliva are detected by a change in the cyclic voltammetry (CV) signal of p-benzoquinone that performs as a reduction-oxidation (RedOx) pH indicator. Our carbon paper electrodes' very high surface area (CPE) is combined with the intrinsic CV fast detection turnover, sensitivity, selectivity, and enzymatic signal amplification to provide fast and effective detection of viral infection within 1 min, directly from unprocessed saliva samples.

## Results
### Sensor design and mode of action
The CPE is fabricated from a conductive carbon paper that contains multi-layers of micro-carbon-fibers (μCF) as a 3D matrix with an ultra-high surface area of 1000–2500 $m^2 g^{-1}$[50,59,60]. Carbon is an attractive material for electrochemical-based sensor development, owing to the well-known chemistry[61], high conductivity, relatively low background currents, and high analytical signal[62,63]. 1 cm$^2$ CPE weighs only about 12 mg; therefore, our CPE's detection window of 0.13 cm$^2$ displays a ca. 3.90 m$^2$ of active working electrode area. Our electrode design and scanning electron microscopy (SEM) images of the μCF are shown in Fig. 1a. As illustrated in Fig. 1b, 3CL$^{pro}$ is targeted specifically by implementing a surface-embedded specific antibody. The 3CL$^{pro}$-specific antibody was drop-casted and physically adsorbed onto the CPE surface. Our modification process is very simple, relies on a single antibody, and requires only two soaking steps with no covalent modification steps required.

Next, a RedOx reactive pH indicator is used to electrochemically detect the pH change brought by its substrate's surface-bound 3CL$^{pro}$ enzymatic hydrolysis, as schematically illustrated in Fig. 1d. The diagnostic signal is amplified by relying on the enzymatic activity turnover rate. Each protease molecule performs hydrolysis of -60 substrate molecules per minute[64], resulting in signal amplification of at least 120-fold within 2 min. A library of substrates has been recognized for 3CL$^{pro}$; the one used in this work showed a high affinity and turnover rate[65–68].

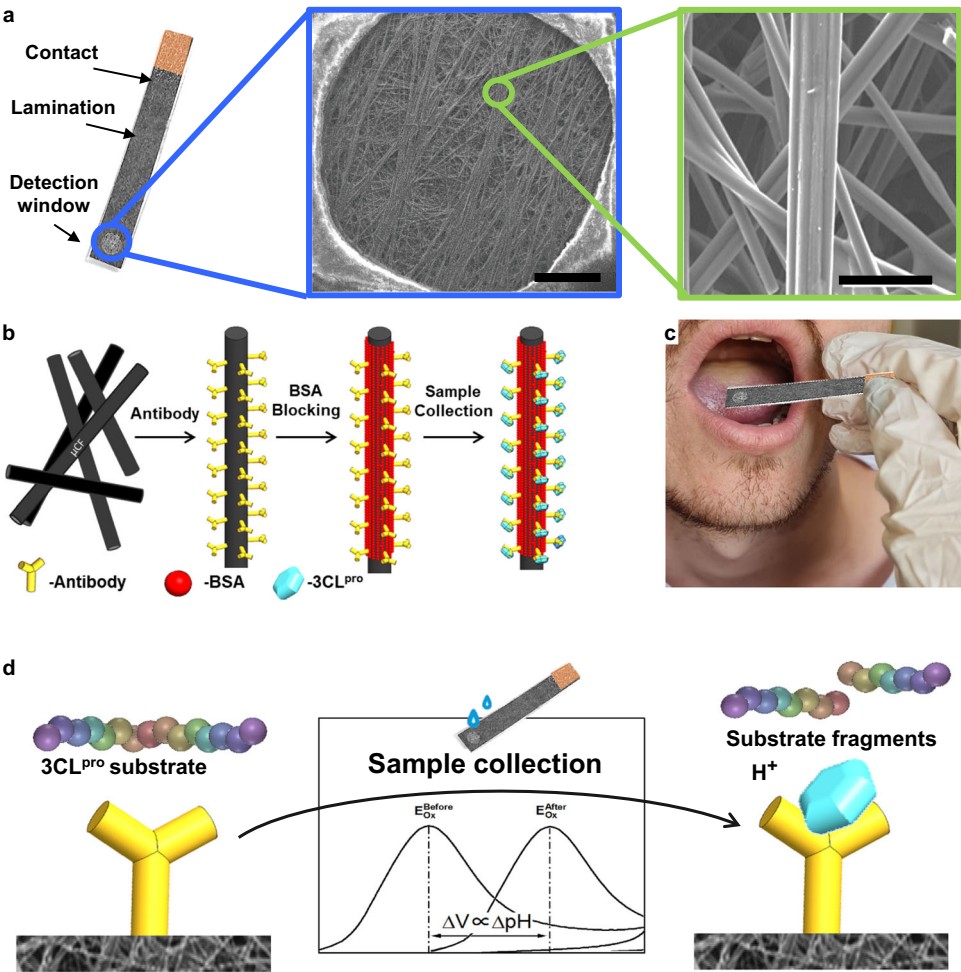

**Fig. 1 | CPE surface, immuno-functionalization, and biosensor method. a** Photo of CPE. Blue inset: SEM images of the detection window, scale bar: 1 mm. Green inset: SEM images of 3D microfiber matrix of CPE, scale bar: 50 μm. **b** Schematic of CPE immuno-functionalization. **c** Illustration of saliva sampling by oral cavity swabbing with CPE. **d** Schematic illustration of the SARS-CoV-2 biosensor detection method.

## 3CL$^{pro}$ activity and RedOx pH indicator characterization

To quantify the pH change brought by the proteolytic activity of 3CL$^{pro}$, 8-Hydroxypyrene-1,3,6-trisulfonic acid (HPTS) was used as a fluorogenic pH indicator[69]. The pH measurements of 80 pmol 3CL$^{pro}$ activity in the presence of 8 nmol 3CL$^{pro}$ substrate are shown in Fig. 2a. For the HPTS-related pH fluorescence response calibration curve, see Supplementary Fig. 1.

The pH drops 0.63 units within two minutes since the theoretical pKa of the substrate peptide fragment is about 2.5; this indicates that about 90 pmol of the substrate was enzymatically cleaved. For a cell volume of 900 μl, the same amount of 3CL$^{pro}$ would theoretically yield a pH change of ~0.74 units; see complete calculation in Supplementary Discussion. Notably, the pH plateaus at 8 min, despite excess substrate available. 3CL$^{pro}$ goes through 3D structure changes that have been reported with pH changes in the window from 7.6 to 6.0[70], possibly affecting enzymatic efficiency, which is reported to be maximal around pH 7[71]. Therefore, 3CL$^{pro}$ activity is self-limiting, with pH being an in vitro stop-signal, and we can infer that there is no necessity for incubation time longer than two minutes under these conditions.

Next, we make use of a RedOx reactive pH indicator able to indicate the expected pH change in the active enzymatic range. In this context, several quinones have been shown to change their electrochemical RedOx potential under different pH environments[72,73]. In this work, *p*-benzoquinone (pBQ) has been chosen as a pH-dependent RedOx probe. pBQ undergoes a two-electron reduction reaction,

accompanied by a reaction with up to two protons (2e$^-$/2H$^+$), depending on the solution pH, see Fig. 2b. The RedOx peaks shift (pH lower than 10) is derived from Eq. (1)[72–75]:

$$E^0_{pH} = E^0_{pH7} - \frac{2.3RT}{nF} m \times pH \qquad (1)$$

Where $E^0$ is the reaction standard potential, $R$ is the universal gas constant, $T$ is the temperature in Kelvin, $n$ is the number of electrons transferred, $F$ is the Faraday constant, and $m$ is the number of protons transferred. When $n = 2$, $m = 2$, and $T = 298$ K, the potential change expected per pH unit is[76]:

$$\frac{\partial E^0_{pH}}{\partial pH} = \frac{2.3RT}{F} \approx 60 \frac{mV}{pH} \qquad (2)$$

These peak shifts are evident in Fig. 2c and linearly plotted in Fig. 2d. While both plots fit well with the linear trend, $R^2 > 0.9$, oxidation peak shifts showed better fittings and more significant peak potential shifts. Consequently, the oxidation peak shift was chosen as the detection marker. The peak shift as a response to 3CL$^{pro}$ activity has been calculated using Eq. (3):

$$\Delta E_{Peak} = E^{Substrate}_{Peak} - E^{Sample}_{Peak} \qquad (3)$$

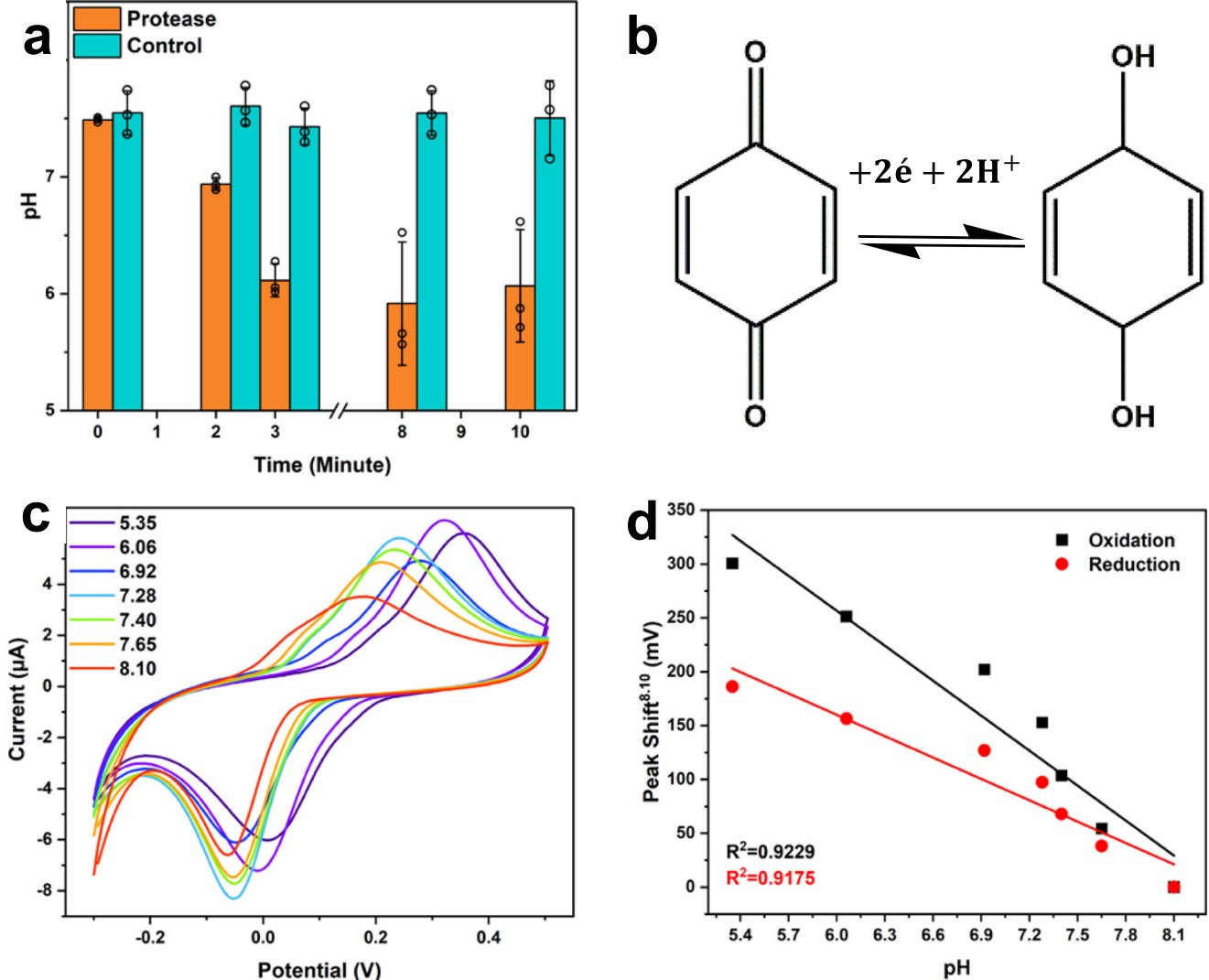

**Fig. 2 | Characterization of pBQ as a RedOx pH indicator. a** Measured pH change caused by 3CL$^{pro}$ (1 μM) activity in the presence of 3CL$^{pro}$ substrate (100 μM, orange plot) and the absence of 3CL$^{pro}$ substrate (blue plot), columns represent mean ± SD from three technical repetitions. **b** pBQ RedOx reaction mechanisms, depending on pH. **c** CV curves of pBQ (15 μM) in PB (900 μl, 10 mM), NaCl (75 mM), pH 5.35–8.10. Scan rate: 0.1 V s$^{-1}$, vs. Ag/AgCl. **d** Linear plots of shifts in the potential of CV peaks of oxidation (black) and reduction (red) from values measured at pH 8.10.

Where $E_{Peak}^{Sample}$ is the voltage at maximal oxidation current after 2 min of CPE incubation in the sample, and $E_{Peak}^{Substrate}$ is the voltage at maximal oxidation current after adding 3CL$^{pro}$ substrate.

**Immuno-functionalization of sensor's surface**

First, protein permeation through the CPE's μCF matrix has been tested by confocal fluorescence microscopy of a bare CPE, Fig. 3a, compared to GFP-treated CPE, Fig. 3b, showing protein permeating through the full depth of the μCF matrix. See Supplementary Fig. 3 for the fluorescence intensity curves of these electrodes.

When combined with strong electrostatic attraction, proteins quickly and strongly adsorb to the CPE surface, as evident in the antibody adsorption plot, shown in Fig. 3c. The enormous surface area of the CPE allows for a very high antibody density per geometric area compared to a planar surface. The results in Fig. 3c indicate the adsorption of $2.0 \times 10^{14}$ antibody molecules per cm$^2$ to the CPE after only 10 min of incubation. This method of immuno-functionalization is possible due to the very strong physical attraction of the antibody molecules to the carbon surface and does not rely on multiple reaction steps, contrary to surface covalent immobilization strategies. The strong bonds

created between the antibody molecules and the CPE surface showed to be highly stable, Fig. 3d, with <10% of the antibody molecules adsorbed to the surface desorbing after a period of 2 h. On the other hand, the advantage of a highly adsorbent large surface matrix may pose a considerable challenge for specific biosensors. Thus, a simple and rapid blocking step by soaking the antibody-modified CPE in a bovine serum albumin (BSA) solution is required (or skimmed milk).

The immuno-functionalization of the CPE surface has been confirmed by high-resolution SEM imaging, as the edge of a bare μCF shown in Fig. 3e has clearly been coated by an organic matter after antibody drop-casting, Fig. 3f. Accordingly, atomic concentration results from Energy-dispersive X-ray spectroscopy (EDS) and X-Ray Photoelectron Spectroscopy (XPS) measurements, Fig. 3g, h, respectively, show an evident rise in the Nitrogen atomic content as a result of adsorption of antibody molecules. EDS and XPS spectra are shown in Supplementary Fig. 4.

Electrochemical CV measurements of [Fe(CN)$_6$]$^{3-}$/[Fe(CN)$_6$]$^{4-}$ confirm the formation of the antibody recognition layer on the surface of the electrode, hindering electron transfer to the negatively-charged RedOx agent[77], Fig. 3i. Additional electrochemical CV measurements of

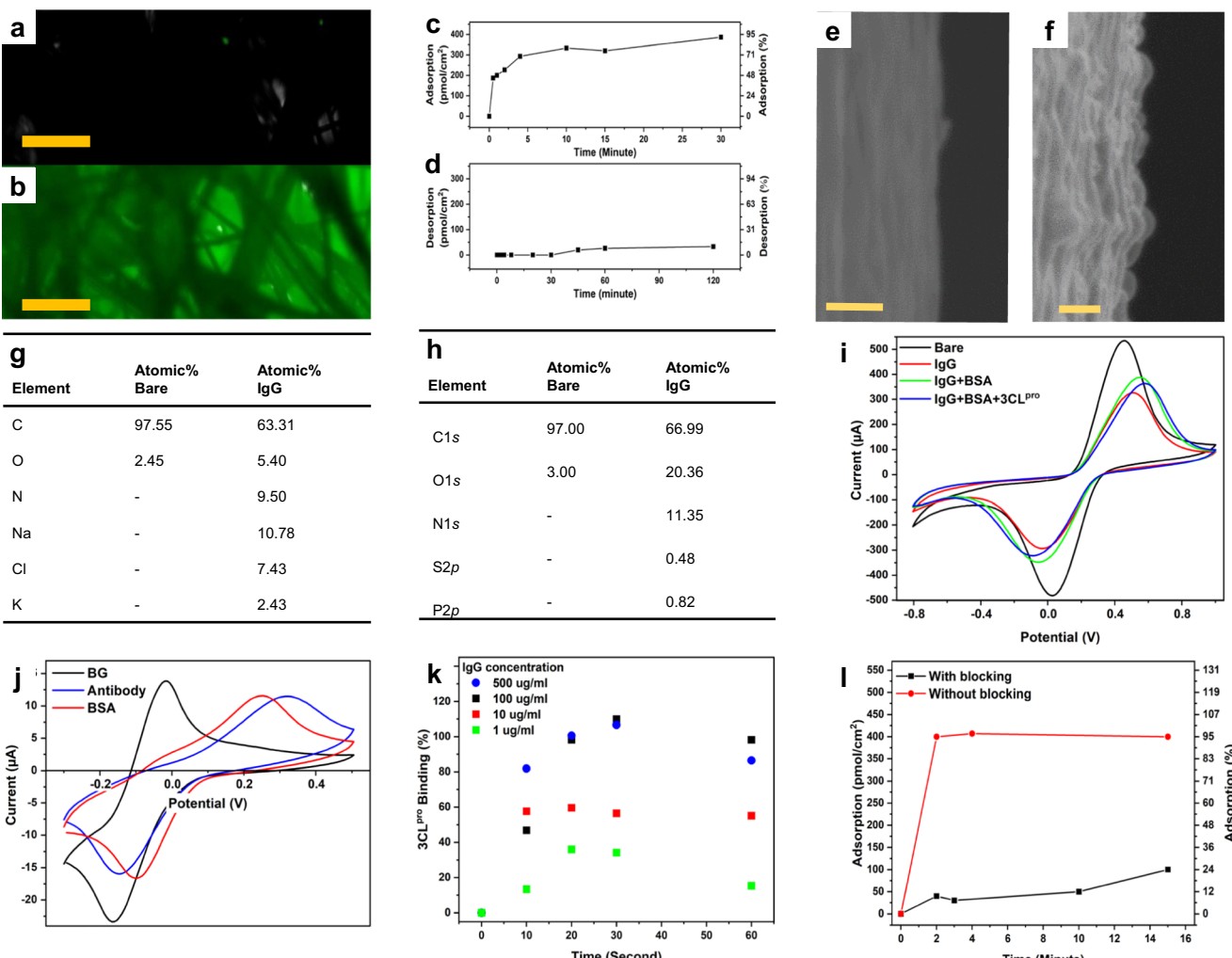

**Fig. 3 | CPE immuno-functionalization. a** Fluorescence microscopy image of bare CPE measured in PBS, scale bar: 1 mm. **b** Fluorescence microscopy image of CPE treated with GFP measured in PBS, scale bar: 1 mm. **c** Antibody adsorption curve of 3CL$^{pro}$-specific antibody onto clean CPE. **d** Antibody desorption curve of 3CL$^{pro}$ antibody from 3CL$^{pro}$ antibody-treated CPE. **e** HR-SEM image of surfaces of bare CPE, scale bar: 100 nm. **f** HR-SEM image of surfaces of CPE treated with 3CL$^{pro}$ antibody, scale bar: 100 nm. **g** Atomic concentrations percentage results from EDS measurements before and after immuno-functionalization. **h** Atomic concentrations percentage results from XPS measurements before and after immuno-functionalization. **i** CV curves of immuno-functionalization steps, Untreated (black) treated with 3CL$^{pro}$ antibody (red), BSA (green), and after 2-min exposure to 3CL$^{pro}$ (blue). CV curves were obtained in 900 µl of 10 mM [Fe(CN)$_6$]$^{3-}$/[Fe(CN)$_6$]$^{4-}$ (1:1), 0.1 M PB, 0.1 M NaCl, pH 7.0, scan rate 0.1 V s$^{-1}$, vs. Ag/AgCl. **j** CV curves of treatment steps, Untreated (black) treated with 3CL$^{pro}$ antibody (red) and BSA (blue). CV curves were obtained in 900 µl of 15 µM pBQ, 25 µM PB, 75 mM NaCl, pH 7.4, scan rate 0.1 V s$^{-1}$, vs. Ag/AgCl. **k** Specific protein-binding curves of 1–500 µg ml$^{-1}$ 3CL$^{pro}$ onto CPE treated with specific 3CL$^{pro}$ IgG and BSA. **l** Non-specific protein-binding curve of CA-15.3 onto CPE treated with a 3CL$^{pro}$-specific antibody with (black curve) and without BSA blocking (red curve).

the redox marker pBQ through CPE immuno-functionalization steps can be found in Fig. 3j.

The 3CL$^{pro}$ specific-binding plot is shown in Fig. 3k, with different curves representing different antibody densities on the CPE surface. Remarkably, maximal binding is reached after only 20 s of incubation in a 3CL$^{pro}$ for the highest antibody surface density tested, which opens the possibility of shortening sample incubation time even further. Furthermore, the physical adsorption of the antibody for 3CL$^{pro}$ to the ultra-high surface of the CPE 3D matrix has been shown to be highly efficient both in terms of kinetics (few minutes) and in retention of antibody affinity to 3CL$^{pro}$, as specific-binding of 3CL$^{pro}$ is efficient even after antibody physical adsorption to the CPE.

Salivary and nasal fluids contain many proteases that aid food disassembly and protect against infections. Proteomic analysis of human saliva has been recognized as a reliable non-invasive alternative to blood testing for diagnostics and disease monitoring[78], including SARS-CoV-2[79]. However, highly active and concentrated proteases are present in saliva, potentially hindering the proteomic salivary diagnostics[80] and expected to affect 3CL$^{pro}$ detection, though 3CL$^{pro}$ has no human homolog[27]. These proteases could potentially cleave our chosen 3CL$^{pro}$ substrate. Therefore, there is a need for specific 'fishing' of 3CL$^{pro}$ out of saliva. In order to prevent non-specific adsorption of the many species found in saliva, the CPE was soaked in BSA. BSA adsorbs to available open sites and prevents further non-specific adsorption of undesired components in the tested saliva samples. The effectiveness of this step is shown in Fig. 3l. Following bio-functionalization, the electrode is exposed to a saliva sample for two minutes; in this step, 3CL$^{pro}$ found in the saliva sample specifically binds to the surface-embedded antibody molecule.

## Optimization of 3CL$^{pro}$ protein detection

For 80 pmol of 3CL$^{pro}$, the measured pH change was 0.63 units, calculated to be about 0.74 units in a cell of 900 µl, which is expected to induce a peak shift of ca. 44 mV. Measurements correlate to this

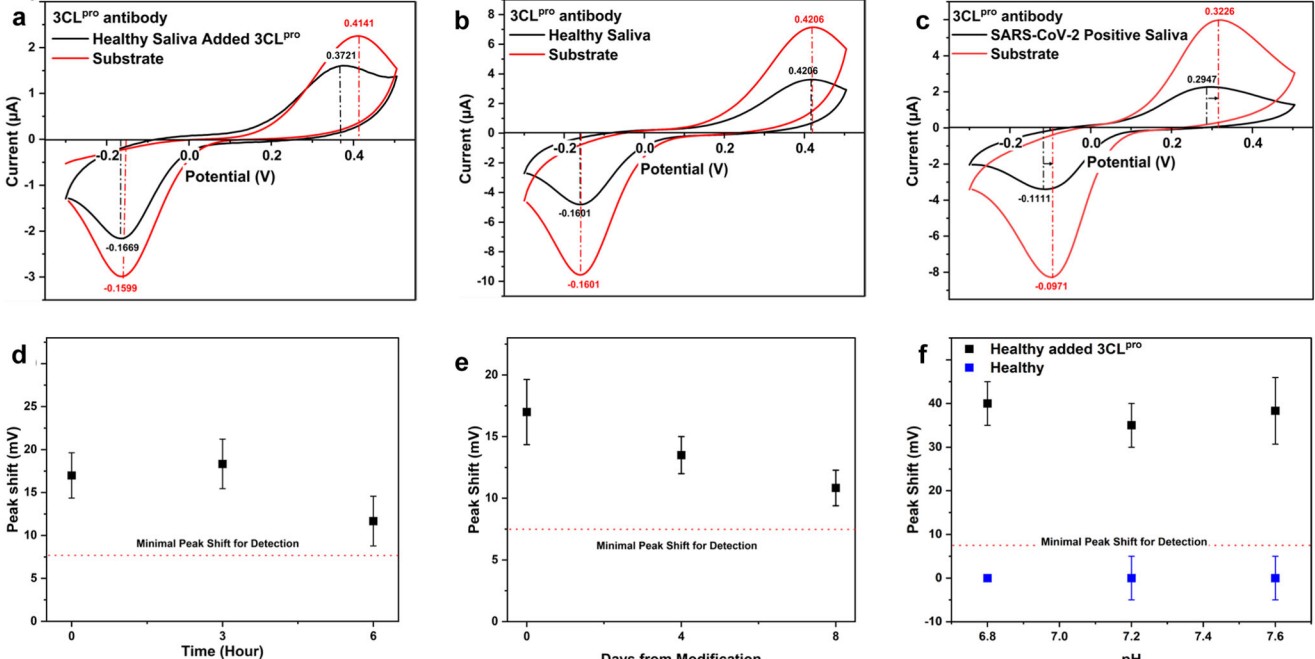

**Fig. 4 | CV performance of CPE for SARS-CoV-2 detection. a** CV curves of CPE treated with 3CL$^{pro}$ antibody and exposed to SARS-CoV-2 negative saliva spiked with 0.2 pmol 3CL$^{pro}$ before (black) and after (red) exposure to 3CL$^{pro}$ substrate. **b** CV curves of CPE treated with 3CL$^{pro}$ antibody and exposed to SARS-CoV-2 negative saliva before (black) and after (red) exposure to 3CL$^{pro}$ substrate. **c** CV curves of CPE treated with 3CL$^{pro}$ antibody and exposed to PCR SARS-CoV-2 positive saliva, before (black) and after (red) exposure to 3CL$^{pro}$ substrate. **d** pBQ oxidation peak shifts of healthy saliva spiked with SARS-CoV-2 3CL$^{pro}$ 50 μg ml$^{-1}$ measured at different times from saliva spiking. After spiking, the spiked saliva

sample was stored at 4 °C. **e** pBQ oxidation peak shifts of healthy saliva spiked with SARS-CoV-2 3CL$^{pro}$ 50 μg ml$^{-1}$ were measured at different times from CPE immuno-functionalization. After immuno-functionalization, CPEs were stored at 4 °C. **f** pBQ oxidation peak shifts of healthy saliva (blue curve), and healthy saliva spiked with 3CL$^{pro}$ 80 μg ml$^{-1}$ (black curve) from different individuals with different initial salivary pH. All CV curves were obtained in 900 μl of 15 μM pBQ, 25 μM PB, 75 mM NaCl, pH 7.4, scan rate 0.1 V s$^{-1}$, vs Ag/AgCl. For **d**, **e**, **f**, data points represent mean ± SD from three technical repetitions.

calculation nicely. CV results of healthy saliva 'spiked' with 58 pmol 3CL$^{pro}$, which were expected to produce a response of $\Delta E_{Peak}$ = ~ 36 mV, show an oxidation peak shift of 38 mV, see Fig. 4a. Importantly, Fig. 4a also indicates that enzyme-antibody binding does not affect the 3CL$^{pro}$ enzymatic activity. This was expected since the chosen antibody targets amino acids 81–132 in 3CL$^{pro}$, and the catalytic dyad is Cys145-His41[81]. Also, while SARS-CoV-2 3CL$^{pro}$ is considered highly conserved, sharing 96.08% sequence identity with SARS-CoV 3CL$^{pro}$ and 87.00% with 3CL$^{pro}$ from the middle east respiratory syndrome coronavirus (MERS-CoV)[81], sequence changes could be used to ensure antibody specificity.

After confirming the expected response in saliva containing 3CL$^{pro}$, non-specific adsorption of salivary proteases was tested by measuring the response to SARS-CoV-2 PCR-negative saliva. A non-specific response is not observed in CV measurements of saliva from healthy individuals without added 3CL$^{pro}$, Fig. 4b, confirming specific 'fishing' of the 3CL$^{pro}$ enzymatic biomarker.

To confirm the necessity of pBQ for 3CL$^{pro}$ detection, a measurement of a 3CL$^{pro}$ solution and its substrate in the absence of pBQ, Supplementary Fig. 5, showed no peaks, thus not allowing for peak shift detection. In addition, CV measurements of healthy participants' saliva spiked with 3CL$^{pro}$, using CPE functionalized with myoglobin-specific antibody, that is non-specific to 3CL$^{pro}$, Supplementary Fig. 6, showed no response with $\Delta E_{Peak}$ = 0 mV, further emphasizing the immuno-functionalization specificity.

CV measurements showing 3CL$^{pro}$ activity positively detected in a saliva sample from PCR-positive to SARS-CoV-2 participant (25 < Ct < 31), Fig. 4c, is remarkably similar to CV results of saliva 'spiked' with 3CL$^{pro}$, Fig. 4a. This confirms the real-world applicability of our platform for SARS-CoV-2 detection, with a clear response of $\Delta E_{Peak}$ = 28 mV, indicating the presence of 3CL$^{pro}$ in the sample and

enabling the detection of SARS-CoV-2 directly from untreated saliva samples.

To test the stability of 3CL$^{pro}$ enzyme in saliva, a sample from a healthy subject was spiked with 50 μg ml$^{-1}$ 3CL$^{pro}$ and tested at different time points. The results in Fig. 4d show that 3CL$^{pro}$ is still active after 6 h in saliva. In addition, the stability of CPE immuno-functionalization over time was tested by storing immune-functionalized CPEs under refrigeration, followed by testing 3CL$^{pro}$-spiked saliva at different times, Fig. 4e. These measurements showed that immuno-functionalization remains active for 8 days under refrigeration at 4–8 degrees celcius.

As saliva pH variability could be expected in broad screenings tests [82], samples of different initial salivary pH were tested, Fig. 4f, to reveal that the 3CL$^{pro}$ specific fishing by its antibody, and electrode washing, effectively prevents initial salivary pH from affecting peak shift results.

## Detection of SARS-CoV-2 in clinical samples

To validate the clinical detection of SARS-CoV-2 in real saliva samples, a set of twenty-four SARS-CoV-2 negative samples (i.e., healthy) and twenty-six SARS-CoV-2 positive samples (PCR-positive, 25 < Ct < 31) were tested. Out of twenty-six SARS-CoV-2 positive samples, all have been positively detected and easily differentiated from healthy samples, since the mean peak shift of SARS-CoV-2 positive samples is ca. 20 mV, Fig. 5a, while healthy samples' mean peak shift is ca. 0.35 mV.

Significant oxidation-peak shift differences were also evident between measurements of samples from SARS-CoV-2 positive participants, and COVID-19 recovered patients, as summarized in Fig. 5b. These results confirm no lingering 3CLpro activity occurs after viral infection ceases.

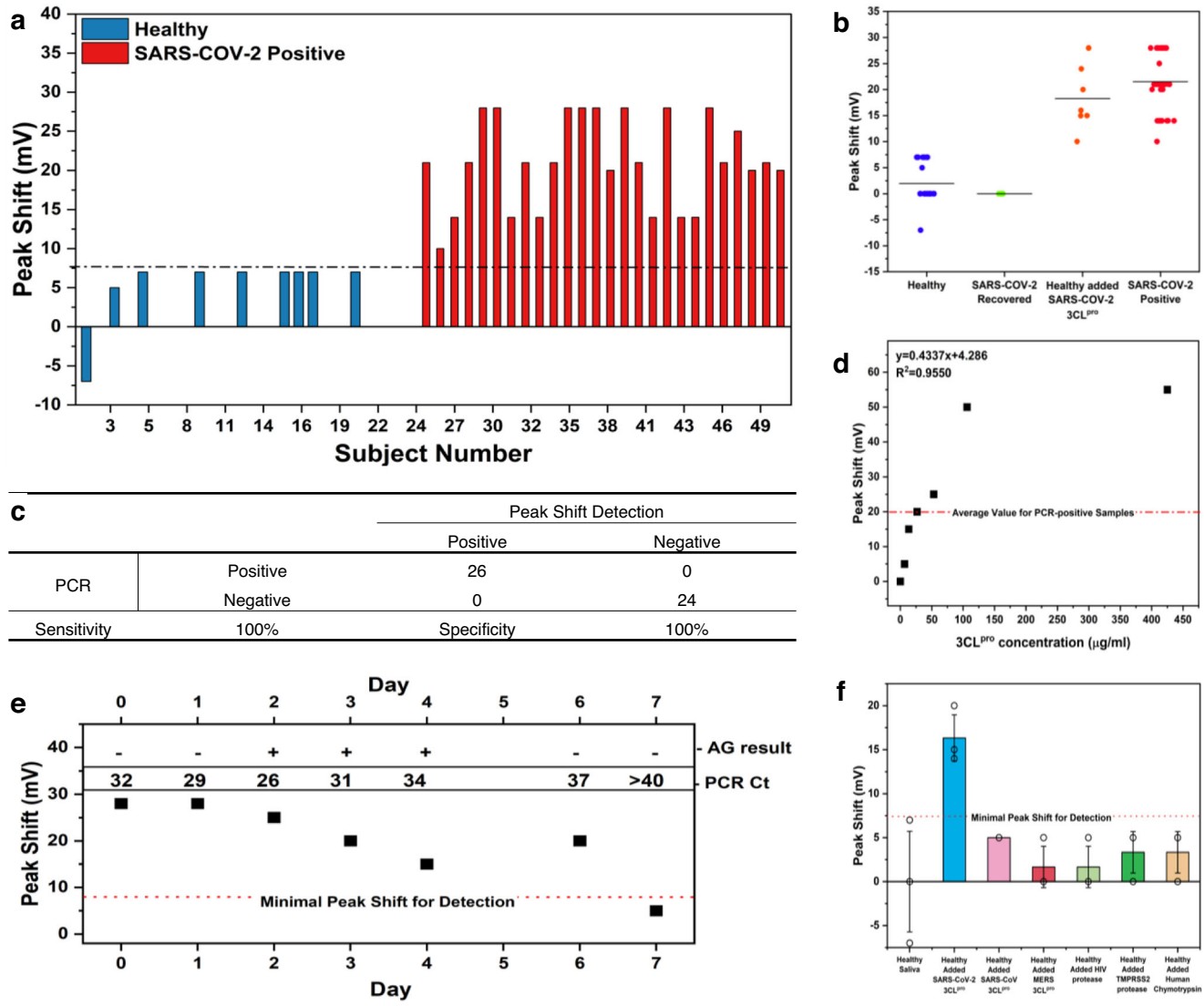

**Fig. 5 | SARS-CoV-2 detection in clinical samples. a** pBQ oxidation peak shift of healthy (blue) and SARS-CoV-2 PCR-positive (red) saliva samples. **b** Scatter plot of pBQ oxidation peak shift of SARS-CoV-2 negative saliva (blue, $N = 24$), the saliva of recovered COVID-19 patients (green, $N = 4$), SARS-CoV-2 negative saliva spiked with $3CL^{pro}$ (orange, $N = 7$), and PCR-positive SARS-CoV-2 (red, $N = 26$) saliva samples. Horizontal lines represent mean peak shift values. **c** Sensitivity/specificity table. **d** Peak shift vs $3CL^{pro}$ concentration. **e** Peak shifts over time from infection of one individual with respect to PCR and antigen test results. **f** Peak shifts of saliva spiked with different proteases. Columns represent mean ± SD from three distinct biological replicates.

Specificity and sensitivity values calculated relatively to PCR results are shown in Fig. 5c, and a plot of peak shift vs $3CL^{pro}$ concentration is shown in Fig. 5d. Both indicate promising attributes for our detection method, with 100% specificity and 100% sensitivity, and a LOD = 6.6 μg ml⁻¹. Even though we do not propose this method for viral quantification, but simply for viral detection, $3CL^{pro}$ quantification could be achieved at concentrations from 13 μg ml⁻¹ up to 106 μg ml⁻¹, with $R^2 = 0.935$. This, combined with patients' saliva samples results, indicate that SARS-CoV-2 positive saliva samples contain 14–59 μg ml⁻¹ of $3CL^{pro}$.

Infection kinetics of one PCR-positive individual has been measured for 8 days since the onset of mild symptoms and compared to PCR Ct values and COVID-19 salivary antigen home-detection kit results, Fig. 5e. Peak shift detection correlated well with PCR Ct results, both showing undetectable values by Day 8 post-symptoms onset. COVID-19 salivary antigen home-detection kit results were falsely negative for 2 days after the onset of PCR positive results. In this, we successfully proved that our method is comparable with PCR detection.

Moreover, no detectable responses were observed when measuring healthy saliva samples spiked with other proteases, Fig. 5f.

$3CL^{pro}$ originating from other coronaviruses (SARS-CoV, MERS-CoV), human immunodeficiency virus protease, and the human protease TMPRSS2 have been tested, showing peak shifts were lower than our detection limit. Considering the high similarity shared between $3CL^{pro}$ from SARS-CoV, MERS-CoV, and SARS-CoV-2, these results show high specificity against potential interferents.

Importantly, results are highly reproducible and accurate, as shown in Supplementary Fig. 8, with ten consecutive experiments testing a saliva sample from a SARS-CoV-2 negative subject giving near-identical results, with an oxidation peak shift standard deviation of 2.5 mV.

## Discussion
Though highly accurate SARS-CoV-2 detection is achieved by methods such as RT-PCR, these are unsuitable for POC applications, as their detection turnover rate is exceedingly low, and they require expensive machinery, trained personnel, and multiple expensive and sensitive reagents.

Here, we have successfully proved the potential of our immuno-functionalized 3D conductive electrodes as a platform for the reliable

and ultrafast detection of SARS-CoV-2 directly from saliva samples in under one minute, using a single antibody capturing agent. Preliminary measurements of SARS-CoV-2 positive and healthy saliva samples established our methods' sensitivity and specificity, equivalent to laboratory RT-PCR. The detection based on 3CL[pro] enzymatic activity could potentially be more reliable, as detecting RNA may give false-positive results by detecting viral RNA fragments residues long after the viral replication is no longer active[18].

We have demonstrated the presence and activity of 3CL[pro] in saliva samples of SARS-CoV-2 positive individuals, which could be further detected by a change in the CV characteristics of the RedOx label pBQ. Importantly, this demonstration of the presence of 3CL[pro] in salivary samples from SARS-CoV-2 patients has not been shown previously. All measurements were taken after one minute of exposure to saliva samples. As previously mentioned, this detection time could be further shortened by modulating the surface area, antibody density, and cell volume of the detection set-up to potentially reach detection turnover cycles of several tens of seconds.

As mentioned before, electrochemical biosensors are widely used as POC analytical devices[52,55] with miniaturized instruments as small as a cellphone and ease of use, allowing people to perform experiments under field conditions, without special training[47]. Thus, we believe our method could provide a large-scale, fast, and accurate SARS-CoV-2 detection platform, thus allowing timely implementation of measures to curb pandemic progression.

## Methods

### Ethical statement
The research presented here complies with all relevant ethical regulations of the Tel Aviv University Ethics Committee (IRB protocol number 71.19). Informed consent was obtained from all participants, and participants were not compensated.

### Materials and chemicals
CPE (254 µm thick, type Spectracarb 2050A-1050, Fibers Technology), Ag/AgCl reference electrode (type RE-1CP, ALS Ltd.), Pt electrode (99.999%, 1 mm diameter, Holland-Moran Ltd), 3CL[pro] enzyme (Recombinant derived from *Escherichia coli*, ab277614, ABCAM), 3CL[pro] specific antibody (Rabbit-derived polyclonal, NBP3-07062, Novous Biological), 3CL[pro] substrate (peptide, KTSAVLQSGFRKME, Sigma-Aldrich), GFP (Recombinant derived from *Escherichia coli*, ab84191, ABCAM), myoglobin antibody (Monoclonal rabbit-derived, ab77232, ABCAM), HIV-2 Protease (Recombinant derived from *Escherichia coli*, ab84117, ABCAM), CA-15.3 (Recombinant derived from *Escherichia coli*, ab80082, ABCAM), Human TMPRSS2 protein (Recombinant derived from Wheat germ, ab112364, ABCAM), MERS-CoV 3CL Protease (Recombinant derived from *Escherichia coli*, E-719, Novous Biological), SARS-CoV 3CL Protease (Recombinant derived from *Escherichia coli*, E-718, Novous Biological), Chymotrypsin protein (Native human, ab90927, ABCAM), p-benzoquinone (reagent grade, Sigma-Aldrich), Acetone (9005-68, J.T.Baker), Isopropanol (IPA, 9079-05, J.T.Baker), Deionized water (DIW, 18 MΩ·cm), PBS (40 mM NaCl, 10 mM phosphate buffer, and 3 mM KCl, pH 7.4; Sigma-Aldrich), Disodium hydrogen phosphate (S7907, Sigma-Aldrich), COVID-19 antigen saliva test (P2004s, GenSure). For protein sequences, see Supplementary Table 1.

### Electrode fabrication and immuno-functionalization
Carbon paper was cut into rectangular pieces of 7 × 50 mm, and laminated with polyethylene at 75 °C to prevent solution capillary rising and contact wetting. An active window of 4 mm diameter was designed and left un-laminated out of the CPE.

For immuno-functionalization, CPE was washed with IPA and DIW, and then 2 µl of 3CL[pro] antibody was drop-casted on CPE's active window. CPE was then washed well with PBS, soaked for 20 min in BSA (5 mg ml⁻¹), and washed again with PBS.

### Measurements of enzymatic pH change
Fluorescence of HPTS (excitation 430 nm, emission 470 nm) of 80 µl of HPTS (80 pmol), 3CL[pro] (80 pmol), and 3CL[pro] substrate (8 nmol) in 10 mM disodium hydrogen phosphate pH 7.5 was measured at different times. PH values were calculated using a calibration curve, see Supplementary Fig. 1. Control was 80 µl of HPTS (80 pmol) and 3CL[pro] (80 pmol) in 10 mM disodium hydrogen phosphate pH 7.5.

### Electrochemical measurements
All electrochemical experiments were performed using a potentiostat (EmStat[3], PalmSense) using PSTrace 5.6 Software. CV measured from −0.3 V to 0.5 V, scan rate 0.1 V s⁻¹. A 3-electrode cell was used, with commercial Ag/AgCl as the reference electrode and Pt as a counter electrode. Measurements were conducted in 900 µl of 25 µM PB, 75 mM NaCl, 32 µM 3CL[pro] substrate, and 15 µM p-benzoquinone.

### Imaging
Light microscopy used Olympus BX41m-LED using a U-PMTCV camera adapter in dark-field mode. Scanning electron microscopy (SEM) imaging used Quanta 200FEG ESEM, Thermo Scientific, 20.0 kV, WD 10.0 mm, and high-resolution SEM (HR-SEM) imaging used Gemeni-SEM-300, Zeiss, 0.500 kV, WD 4.6 mm.

### Protein adsorption, desorption, and binding measurements
Proteins were quantified using NanoDrop One/OneC spectrophotometer, Thermo Scientific. For adsorption, CPE was soaked in protein solutions, and the adsorbed amounts at a time point $t_i$ were calculated using Eq. (4):

$$\text{Adsorption}\,(t_i) = \frac{C_0 - C_i}{C_0} \times 100\% \tag{4}$$

Where $C_0$ is the initial concentration, and $C_i$ is the concentration at $t_i$. Binding and desorption were calculated in the same fashion.

### Clinical sample preparation and testing
The human saliva experiments were approved by the Ethics Committee of Tel Aviv University. Saliva samples were collected in sterile 15 ml sterile tubes and kept refrigerated until measuring. No pretreatment steps were taken before measurements.

### Population characteristics
The human participants were female and male adults from the age of 18 years old up to the age of 78 years old. No additional clinical information on donors was collected during the study. SARS-CoV-2-positive participants were recruited after a PCR-positive test. Healthy individuals were recruited and validated as SARS-CoV-2 negative after a PCR test. No self-selection bias or other biases occurred during the study that may have affected the results presented.

### Statistics and reproducibility
No statistical method was used to predetermine the sample size. Each sample test in this manuscript was repeated at least twice. All attempts at replication were successful. PH error bars in Fig. 2a were derived from the SD of technical repetitions ($n = 3$). Peak shift errors were calculated from the SD of technical repetitions ($n = 3$).

### XPS measurements
XPS measurements were performed in UHV ($2.5 \times 10^{-10}$ Torr base pressure) using Thermo Scientific™ Nexsa G2 System.

### Antigen sample testing
GenSure COVID-19 antigen test for saliva and nasal samples was used for self-testing by PCR-positive SARS-CoV-2 volunteers.

**Reporting summary**

Further information on research design is available in the Nature Research Reporting Summary linked to this article.

## Data availability

The data generated in this study are provided in the Source data files. Source data are provided with this paper.

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

## Acknowledgements

This research has received no external funding.

## Author contributions
E.B. performed experimental studies, data analysis, and article drafting and revising. E.G. performed experimental studies. F.P. contributed to the study conception and supervised the work.

## Competing interests
The authors declare no competing interests.
