## [Peer Review File · Nature Communications]

REVIEWER COMMENTS

Reviewer #1 (Remarks to the Author):

In this work, 3CLpro as a specific biomarker of active infection is detected in saliva by a change in the cyclic voltammetry (CV) signal of p-benzoquinone that performs as a reduction-oxidation (RedOx) pH indicator. 3CLpro is captured by its specific antibody on carbon paper electrodes with very high surface area (CPE). The authors talked about the sensitivity and selectivity of the proposed assay, but they did not provide any data to support their claim. What is LOD, LOQ, What is the specificity of detection and what is the signal amplification in the presence of other non-SARS-CoV-2 samples (i.e influenza and Rhinovirus) saliva samples? Human sample could also have different proteinase that could interfere with the signal. Authors should run a control experiment using some other proteinase such as human chymotrypsin and other virus proteinase to see the false positive results and interference.

The functionalization of CPE with antibody and then after exposure to sample (capture 3CLpro) must be characterized by cyclic voltammetry in the presence of i.e. potassium ferrocyanide. The oxidation and reduction peak of potassium ferrocyanide would be changed during functionalization with antibody and in the presence of 3CLpro enzyme. Moreover, AFM or SEM could be used to further validate successful functionalization.

HPTS was used as a fluorogenic pH indicator but there is no data to show its sensitivity and performance. Just the electrochemical oxidation shift of HQ/Q redox probe was chosen as the assay signal. In case of infection, pH of patient sample also change it is also needed to monitor the pH of sample whenever pH change is used as a parameter to get a signal.

Electrochemical measurements were conducted in 900 ml of 25 mM PB, 75 mM NaCl, 32 mM 3CLpro substrate, and 15 mM p-benzoquinone. What is the point of using PB buffer in the measurement signal while the detection is based on pH change? How the pH of sample would be changed in 25mM PB that has a strong buffer capacity?

Figure 3: How the adsorption of antibody and 3CLpro on CPE was measured? As authors reported in line 161 "The 3CLpro-specific antibody was drop-casted and physically adsorbed onto the CPE surface." is this adsorption stable and how strong is the adsorption binding?

When author tested clinical sample and compared with RT-PCR what is the Ct value for RT-PCR positive sample

Reviewer #2 (Remarks to the Author):

The authors have demonstrated that the 3CLpro protease of coronaviruses can be detected using an electrochemical-based assay after capture on a surface from saliva. This approach represents a novel direction for sensitive detection of coronaviruses from clinical samples. However, there remain limitations in the applicability of this work as outlined below which should ideally be addressed prior to publication. Most notably, would when would this assay be able to detect samples, would it be specific to SARS-CoV-2 (or to other coronaviruses as well), and how would it be utilized in clinical practice. In addition, the manuscript would benefit from an additional review for both clarity, grammar, and spelling.

Major Comments

1. Description of the early replication events for coronaviruses is a little misleading. Upon entry into the cell, the replicase genes are translated into polyproteins needing cleaved. It should be made clear that it is only the replicase genes that are initially translated and into polyproteins (rather than simply proteins). (lines 67 – 70)
2. The authors discuss that their assays may work better than existing RNA detection platforms because viral RNA fragments last long after infection, however they never show any kinetics for how long during and/or after infection their assay remains sensitive. Furthermore, they also don't show any data on how long 3CLpro remains active in saliva. Given the importance of binding and detecting 3CLpro, it is important that the authors demonstrate the range and suitability of this technique over times of infection rather than against strictly positive vs negative samples. (lines 297 – 299)
3. In addition to kinetics studies, the authors may also consider increasing their sample size to evaluate specificity since other cysteine-like proteases are present in saliva and it remains unclear whether this assay is specific to SARS-CoV-2 or whether it would give false positives for other common cold coronaviruses (229E, HKU1, OC43, and NL63) which share very similar structures and are likely to be captured by the antibodies described in this study.
4. While the authors do show specificity of their assay for positive versus negative saliva samples (Fig 4), it remains unclear how this proof of concept can be scaled to a rapid test. Do the authors plan to do electrochemical analysis in a laboratory setting on submitted samples or develop a colorimetric component for observing results? Please elaborate in the discussion.

Minor Comments

1. "precence" should be "presence" (line 27)
2. Statement describing "another common approaches" is unclear (lines 38 – 39)
3. "SAR-CoV-2" should be "SARS-CoV-2" (line 48)
4. The authors assert that the presence of 3CLpro indicates active infection, however the protease may be able to persist beyond active replication as demonstrated by the applicability of their assay. (lines 106 – 109)
5. The authors describe "SARS-CoV-2" proteins as being expressed as a single polypeptide chain, but there are two separate replicase polypeptides or polyproteins that are produced during initial translation. (lines 111 – 113)
6. The catalytic dyad should be described as individual residues, the current format in the manuscript "C--H" implies that there are two residues between the catalytic His and Cys residues. In reality, these residues are approximately 100 amino acids apart in the protein primary structure. (line 115)
7. What is COVID-2? Please clarify. (line 255)

Response to comments of Reviewer #1

Thank you for your valuable comments and suggestions. Our response to all the questions raised is as follows:

In this work, 3CL^{pro} as a specific biomarker of active infection is detected in saliva by a change in the cyclic voltammetry (CV) signal of p-benzoquinone that performs as a reduction-oxidation (RedOx) pH indicator. 3CL^{pro} is captured by its specific antibody on carbon paper electrodes with very high surface area (CPE). The authors talked about the sensitivity and selectivity of the proposed assay, but they did not provide any data to support their claim. What is LOD, LOQ, What is the specificity of detection and what is the signal amplification in the presence of other non-SARS-CoV-2 samples (i.e influenza and Rhinovirus) saliva samples? Human sample could also have different proteinase that could interfere with the signal. Authors should run a control experiment using some other proteinase such as human chymotrypsin and other virus proteinase to see the false positive results and interference.

Response:

Thank you for the very astute observation that helped strengthen of our work. Experiment results added to the paper in **Figure 5**, indicate promising attributes for our detection method, with 100% specificity and 100% sensitivity, LOD=6.6 $\mu\text{g ml}^{-1}$. Even though we do not intend this method for viral quantification, but simply viral detection, 3CL^{pro} quantification could be achieved at concentrations from 13 $\mu\text{g ml}^{-1}$ up to 106 $\mu\text{g ml}^{-1}$, with $R^2=0.935$. This, combined with patients' results, could indicate that SARS-CoV-2 positive saliva samples contain 14-59 $\mu\text{g ml}^{-1}$ of 3CL^{pro}.

Moreover, no detectable responses were observed when measuring healthy saliva spiked with other proteases, **Figure 5f**. 3CL^{pro} originating from other coronaviruses (SARS-CoV, MERS-CoV), human immunodeficiency virus protease, and the human protease TMPRSS2 have been tested, showing peak shifts were lower than our detection limit. Considering the high similarity shared between 3CL^{pro} from SARS-CoV, MERS-CoV, and SARS-CoV-2, these results show high specificity against potential interferents.

The effect of salivary proteinases has shown to be negligible by testing twenty-four healthy samples with no false positive events.

Figure 5 | SARS-CoV-2 detection in clinical samples. **a.** pBQ oxidation peak shift of healthy (blue) and SARS-CoV-2 PCR-positive (red) saliva samples. **b.** Scatter plot of pBQ oxidation peak shift of SARS-CoV-2 negative saliva (blue), the saliva of recovered COVID19 patients (green), SARS-CoV-2 negative saliva spiked with 3CL^{pro} (orange), and PCR positive SARS-CoV-2 (red) saliva samples. Horizontal lines represent mean peak shift values. **c.** Sensitivity/specificity table. **d.** Peak shifts over time from infection of one individual with respect to PCR and antigen test results. **e.** Peak shift vs. 3CL^{pro} concentration. **f.** Peak shifts of saliva spiked with different proteases. **g.** pBQ oxidation peak shift results of 10 consecutive experiments measuring the same healthy saliva sample, mean value of measurements of PCR SARS-CoV-2 positive saliva samples is on the right for comparison.

The functionalization of CPE with antibody and then after exposure to sample (capture 3CLpro) must be characterized by cyclic voltammetry in the presence of i.e. potassium ferrocyanide. The oxidation and reduction peak of potassium ferrocyanide would be changed during functionalization with antibody and in the presence of 3Clpro enzyme. Moreover, AFM or SEM could be used to further validate successful functionalization.

Response:

We thank the reviewer for this important comment. SEM, XPS, EDS and ferrocyanide CV characterization of the modification have been added to the manuscript.

Figure 3 | CPE immuno-functionalization. **a.** Fluorescence microscopy image of bare CPE measured in PBS, scale bar: 1 mm. **b.** Fluorescence microscopy image of CPE treated with GFP measured in PBS, scale bar: 1 mm **c.** Antibody adsorption curve of 3CL^{pro}-specific antibody onto clean CPE. **d.** Antibody desorption curve of 3CL^{pro} antibody from 3CL^{pro} antibody-treated CPE. **e.** HR-SEM image of surfaces of bare CPE, scale bar: 100 nm. **f.** HR-SEM image of surfaces of CPE treated with 3CL^{pro} antibody, scale bar: 100 nm. **g.** Atomic concentrations percentage results from EDS measurements before and after immuno-functionalization. **h.** Atomic concentrations percentage results from XPS measurements before and after immuno-functionalization. **i.** CV curves of immuno-functionalization steps, Untreated (black) treated with 3CL^{pro} antibody (red), BSA (green), and after 2-minute exposure to 3CL^{pro} (blue). CV curves were obtained in 900 μ l of 10 mM $[\text{Fe}(\text{CN})_6]^{3-}/[\text{Fe}(\text{CN})_6]^{4-}$ (1:1), 0.1 M PB, 0.1 M NaCl, pH 7.0, scan rate 0.1 V sec⁻¹, vs. Ag/AgCl. Antibody adsorption curve of 3CL^{pro} specific IgG antibody onto clean CPE. **j.** CV curves of treatment steps, Untreated (black) treated with 3CL^{pro} antibody (red) and BSA (blue). CV curves were obtained in 900 μ l of 15 μ M pBQ, 25 μ M PB, 75mM NaCl, pH 7.4, scan rate 0.1 V sec⁻¹, vs. Ag/AgCl. **k.** Specific protein-binding curves of 1-500 μ g ml⁻¹ 3CL^{pro} onto CPE treated with specific 3CL^{pro} IgG and BSA. **l.** Non-specific protein binding curve of CA-15.3 onto CPE treated with a 3CL^{pro}-specific antibody with (black curve) and without BSA blocking (red curve).

HPTS was used as a fluorogenic pH indicator but there is no data to show its sensitivity and performance. Just the electrochemical oxidation shift of HQ/Q redox probe was chosen as the assay signal. In case of infection, pH of patient sample also change it is also needed to monitor the pH of sample whenever pH change is used as a parameter to get a signal.

Response:

HPTS was only used to illustrate the pH change caused by the enzymatic activity of 3CL^{pro}, a curve of HPTS fluorescence response to pH is shown in **Supplementary Fig.1**.

As for salivary pH, our platform is insensitive to initial saliva pH since the specific antibody captures 3CL^{pro} from saliva, the electrode is then washed and all measurements are performed at pH 7.4. We have confirmed this claim by measuring the peak shifts of different salivary pH, now in **Figure 4f**.

Electrochemical measurements were conducted in 900 ml of 25 mM PB, 75 mM NaCl, 32 mM 3CLpro substrate, and 15 mM p-benzoquinone. What is the point of using PB buffer

in the measurement signal while the detection is based on pH change? How the pH of sample would be changed in 25mM PB that has a strong buffer capacity?

Response:

We thank the reviewer immensely; this is of course a typo, and the correct buffer concentration is 25 μ M.

Figure 3: How the adsorption of antibody and 3CLpro on CPE was measured? As authors reported in line 161 “The 3CLpro-specific antibody was drop-casted and physically adsorbed onto the CPE surface.” is this adsorption stable and how strong is the adsorption binding?

Response:

Proteins were quantified using NanoDrop One/OneC spectrophotometer, Thermo Scientific. For adsorption, CPE was soaked in protein solutions, and the adsorbed amounts at a time point t_i were calculated using the equation:

$$Adsorption(t_i) = \frac{C_0 - C_i}{C_0} \times 100\%$$

Where C_0 is the initial concentration, and C_i is the concentration at t_i . Binding and desorption were calculated in the same fashion. This explanation has been added under ‘methods’ section. The strong bonds created between the antibody molecules and the CPE surface showed to be highly stable, **Figure 3d**, with less than 10% of the antibody molecules adsorbed to the surface desorbing after a period of 2 hours.

Figure 3 | CPE immuno-functionalization. c. Antibody adsorption curve of 3CL^{pro}-specific antibody onto clean CPE. **d.** Antibody desorption curve of 3CL^{pro} antibody from 3CL^{pro} antibody-treated CPE.

When author tested clinical sample and compared with RT-PCR what is the Ct value for RT-PCR positive sample.

Response:

Ct values for positive samples were between 25 to 37. This information has been added to the text as well.

Response to comments of Reviewer #2

Thank you for your valuable comments and suggestions. Our response to all the questions raised is as follows:

The authors have demonstrated that the 3CLpro protease of coronaviruses can be detected using an electrochemical-based assay after capture on a surface from saliva. This approach represents a novel direction for sensitive detection of coronaviruses from clinical samples. However, there remain limitations in the applicability of this work as outlined below which should ideally be addressed prior to publication. Most notably, would when would this assay be able to detect samples, would it be specific to SARS-CoV-2 (or to other coronaviruses as well), and how would it be utilized in clinical practice. In addition, the manuscript would benefit from an additional review for both clarity, grammar, and spelling.

Major Comments

1. **Description of the early replication events for coronaviruses is a little misleading. Upon entry into the cell, the replicase genes are translated into polyproteins needing cleaved. It should be made clear that it is only the replicase genes that are initially translated and into polyproteins (rather than simply proteins). (lines 67 – 70)**

We thank the reviewer for helping us avoid unclear descriptions. The sentences have been edited.

2. **The authors discuss that their assays may work better than existing RNA detection platforms because viral RNA fragments last long after infection, however they never show any kinetics for how long during and/or after infection their assay remains sensitive. Furthermore, they also don't show any data on how long 3CLpro remains active in saliva. Given the importance of binding and detecting 3CLpro, it is important that the authors demonstrate the range and suitability of this technique over times of infection rather than against strictly positive vs negative samples. (lines 297 – 299)**

Response:

Thank you for the very important remark, we have tested on individual over 8 days since the onset of minor symptoms. Our peak shift-based detection has been compared to PCR and antigen home kit testing. These results are in **Figure 5e**. Peak shift detection correlated incredibly with PCR Ct result, both showing undetectable values by Day 8 post-symptoms onset. COVID-19 salivary antigen home detection kit results were falsely negative for two days after PCR positive results, and our peak shift detection of SARS-CoV-2 gave false-negative results a day earlier than both of the other methods. In this, we successfully proved that our method is comparable with PCR detection.

As for 3CL^{pro} activity in saliva, we tested the activity over 6 hours from saliva spiking, **Figure 4e.**

3. In addition to kinetics studies, the authors may also consider increasing their sample size to evaluate specificity since other cysteine-like proteases are present in saliva and it remains unclear whether this assay is specific to SARS-CoV-2 or whether it would give false positives for other common cold coronaviruses (229E, HKU1, OC43, and NL63) which share very similar structures and are likely to be captured by the antibodies described in this study.

Response:

Sample size has increased from 28 to 50, with twenty-four SARS-CoV-2 negative samples (i.e., healthy) and twenty-six SARS-CoV-2 positive samples (PCR-positive, $25 < Ct < 31$) were tested. Detection of other coronaviruses have been tested by spiking saliva with 3CL^{pro} from SARS-CoV and MERS, that respectively share 96.08% and 87.00% sequence identity with SARS-CoV-2 3CL^{pro}. No false negative

Figure 5 | SARS-CoV-2 detection in clinical samples. **a.** pBQ oxidation peak shift of healthy (blue) and SARS-CoV-2 PCR-positive (red) saliva samples. **b.** Scatter plot of pBQ oxidation peak shift of SARS-CoV-2 negative saliva (blue), the saliva of recovered COVID19 patients (green), SARS-CoV-2 negative saliva spiked with 3CL^{pro} (orange), and PCR positive SARS-CoV-2 (red) saliva samples. Horizontal lines represent mean peak shift values. **c.** Sensitivity/specificity table. **d.** Peak shift vs. 3CL^{pro} concentration. **e.** Peak shifts over time from infection of one individual with respect to PCR and antigen test results. **f.** Peak shifts of saliva spiked with different proteases. **g.** pBQ oxidation peak shift results of 10 consecutive experiments measuring the same healthy saliva sample, mean value of measurements of PCR SARS-CoV-2 positive saliva samples is on the right for comparison.

4. While the authors do show specificity of their assay for positive versus negative saliva samples (Fig 4), it remains unclear how this proof of concept can be scaled to a rapid test. Do the authors plan to do electrochemical analysis in a laboratory setting on submitted samples or develop a colorimetric component for observing results? Please elaborate in the discussion.

Response:

We thank the reviewer for this comment. In general, there are multiple real-world scenarios where a rapid detection is required, such as airports, sport events, public buildings access, malls, schools/universities/colleges, weddings and many more. In such cases there is no restriction to apply the electrochemical set-up as described in the manuscript. Electrochemical set-ups as the

one described are cost-effective and can be readily multiplexed for the simultaneous on-spot seconds-long rapid detection of Covid19 in such cases where a fast detection turnover is highly required. Furthermore, during these pandemic times where each individual is forced to test himself almost every day at home, the use of cost-effective electrochemical hardware, the whole detection system can be built at a cost of less than a few tens of dollars, can be potentially, practically and widely deployed as well.

Indeed, as cleverly suggested by the reviewer, our group is currently developing a colorimetric direct detection assay, free from electrochemical hardware requirements.

Minor Comments

1. “precence” should be “presence” (line 27)

The spelling was corrected.

2. Statement describing “another common approaches” is unclear (lines 38 – 39)

This statement was omitted while shortening the abstract as to adhere to Nature Communications formatting.

3. “SAR-CoV-2” should be “SARS-CoV-2” (line 48)

The spelling was corrected.

4. The authors assert that the presence of 3CL^{pro} indicates active infection, however the protease may be able to persist beyond active replication as demonstrated by the applicability of their assay. (lines 106 – 109)

This statement was changed to “active viral replication” to better reflect the importance of 3CL^{pro} activity.

5. The authors describe “SARS-CoV-2” proteins as being expressed as a single polypeptide chain, but there are two separate replicase polypeptides or polyproteins that are produced during initial translation. (lines 111 – 113)

This statement was changed with the reviewer’s remark in mind.

6. The catalytic dyad should be described as individual residues, the current format in the manuscript “C--H” implies that there are two residues between the catalytic His and Cys residues. In reality, these residues are approximately 100 amino acids apart in the protein primary structure. (line 115)

The amino acid’s numbering has been added to the paragraph describing the catalytic dyad.

7. What is COVID-2? Please clarify. (line 255)

This was changed to “SARS-CoV-2”.

We hope you will find our revised manuscript suitable for publication in *Nature Communications*.

REVIEWER COMMENTS

Reviewer #1 (Remarks to the Author):

The authors have improved from their initial submission by expanding their study to include a basis for viral quantification, measurements to validate successful antibody bio-functionalization, device stability measurements, specificity measurements, and measurements demonstrating their platform's sensitivity over the course of an active infection.

In response to reviewers' comments regarding sensitivity and specificity, the authors performed additional experiments showing that their detection method provided specificity to 3CLpro originating from SARS-CoV-2 with no significant off-target responses from a handful of other viral samples (Figure 5f). Moreover, their additional experiments indicated that their detection method had a LOD of $6.6 \mu\text{g}\cdot\text{ml}^{-1}$ (lines 310-317, Figure 5d). It is worth noting that the captions for Figure 5d and Figure 5e appear to be switched.

In response to reviewers' comments regarding antibody bio-functionalization, the authors provided results from CV characterization that indicate successful bio-functionalization, further supported by orthogonal outputs from SEM imaging, XPS, and EDS (Figure 3). It is unclear how the atomic % content values provided in Figure 3g and Figure 3h result from the spectra provided in Supplementary Figure 3.

In response to reviewers' comments regarding salivary pH, the authors provided results showing that their detection method does not produce false positives for samples from healthy patients with initial salivary pH ranging from 6.8 to 7.6 (Figure 4f). While this result shows insensitivity to initial salivary pH when no peak shift is expected, it still does not directly indicate whether initial salivary pH affects proper detection of 3CLpro in saliva.

In response to reviewers' comments regarding antibody adsorption stability, the authors indicated that less than 10% of the adsorbed antibody molecules desorb over the course of 2 hours (Figure 3d). However, the percentage of adsorption and desorption (right axes) on Figure 3c and Figure 3d correspond to different surface densities at 100%, neither of which agree with the stated total surface density of 2.0×10^{14} molecules/cm² (~ 332 pmol/cm²).

In response to reviewers' comments regarding their method's sensitivity over time, the authors performed additional experiments showing that their method is successful in detecting 3CLpro over the course of an active infection (Figure 5e), correlating very well with PCR detection, therefore showing comparable performance.

In response to reviewers' comments regarding scaling this proof of concept result to a rapid test, the authors only vaguely address this concern in their rebuttal. While the authors claim that "the whole detection system can be built at a cost of less than a few tens of dollars" and "can be potentially, practically and widely deployed", they fail to provide any details in the discussion as to how this can be accomplished and whether it would be advantageous to performing the electrochemical analysis in a laboratory setting.

Reviewer #2 (Remarks to the Author):

The authors have addressed nearly all of my concerns and I have no further recommended changes to suggest.

Response to comments of Reviewer #1

Thank you for your valuable comments and suggestions. Our response to all the questions raised is as follows:

The authors have improved from their initial submission by expanding their study to include a basis for viral quantification, measurements to validate successful antibody bio-functionalization, device stability measurements, specificity measurements, and measurements demonstrating their platform's sensitivity over the course of an active infection.

In response to reviewers' comments regarding sensitivity and specificity, the authors performed additional experiments showing that their detection method provided specificity to 3CLpro originating from SARS-CoV-2 with no significant off-target responses from a handful of other viral samples (Figure 5f). Moreover, their additional experiments indicated that their detection method had a LOD of 6.6 $\mu\text{g}\cdot\text{ml}^{-1}$ (lines 310-317, Figure 5d). It is worth noting that the captions for Figure 5d and Figure 5e appear to be switched.

We thank the reviewer for the kind remarks and correct observation. The figure captions have been corrected as requested.

In response to reviewers' comments regarding antibody bio-functionalization, the authors provided results from CV characterization that indicate successful bio-functionalization, further supported by orthogonal outputs from SEM imaging, XPS, and EDS (Figure 3). It is unclear how the atomic % content values provided in Figure 3g and Figure 3h result from the spectra provided in Supplementary Figure 3.

Atomic percentage was calculated in Thermo Scientific *Avatage* software, using the following equation:

$$C_A = \frac{I_A/S_A}{\sum_n I_n/S_n} \times 100\%$$

Where C_A is the atomic % content of A, I_A is the intensity of A's peak and S_A is the sensitivity of A atom. This explanation has been added to the supporting section describing **Supplementary Figure 3**.

In response to reviewers' comments regarding salivary pH, the authors provided results showing that their detection method does not produce false positives for samples from healthy patients with initial salivary pH ranging from 6.8 to 7.6 (Figure 4f). While this

result shows insensitivity to initial salivary pH when no peak shift is expected, it still does not directly indicate whether initial salivary pH affects proper detection of 3CLpro in saliva.

We thank the reviewer for this comment. Results showing measurements of 3CLpro-spiked saliva samples with different initial pH values have been added to **Figure 4f** as requested.

In response to reviewers' comments regarding antibody adsorption stability, the authors indicated that less than 10% of the adsorbed antibody molecules desorb over the course of 2 hours (Figure 3d). However, the percentage of adsorption and desorption (right axes) on Figure 3c and Figure 3d correspond to different surface densities at 100%, neither of which agree with the stated total surface density of 2.0×10^{14} molecules/cm² (~332 pmol/cm²).

Thank you for this meticulous and analytical observation. The values in pmol/cm² are correct; however, the axis showing percentages in **Figure 3d** has moved, we are unsure how this software artifact happened but this is now corrected thanks to this important comment. The value 2.0×10^{14} molecules/cm², this value correlates to the measured surface density after 10 minutes of incubation, as stated in the manuscript.

In response to reviewers' comments regarding their method's sensitivity over time, the authors performed additional experiments showing that their method is successful in detecting 3CLpro over the course of an active infection (Figure 5e), correlating very well with PCR detection, therefore showing comparable performance.

We thank the reviewer for suggesting this experiment which has improved the presentation of our work tremendously.

In response to reviewers' comments regarding scaling this proof of concept result to a rapid test, the authors only vaguely address this concern in their rebuttal. While the authors claim that "the whole detection system can be built at a cost of less than a few tens of dollars" and "can be potentially, practically and widely deployed", they fail to provide any details in the discussion as to how this can be accomplished and whether it would be advantageous to performing the electrochemical analysis in a laboratory setting.

A paragraph expanding the ability to implement electrochemical biosensors in POC applications has been expanded under the discussion and introduction sections of the manuscript:

Over the past few decades there has been an upsurge in the use of electrochemical biosensors for analysis of food quality,(Ahmed, Zourob, and Tamiya 2016; Zhang, Zuo, and Ye 2015) environmental monitoring,(Krivitsky et al. 2021; Rogers 2006) and clinical diagnostics.(Anon n.d.; Wang 2006) Electrochemical biosensors display higher selectivity and sensitivity, faster response times, and require a lower amount of sample volumes when compared to traditional standard methods.(Bhalla et al. 2020; Lim and Bonanni 2020) These attributes, combined with operational simplicity, cost-effectiveness, multiplexing capability, and possibility for miniaturization, lead to electrochemical biosensors being frequently designed for POC analysis.(Srinivasan and Tung 2015; Wang 2006)

The miniaturization of formerly used complex benchtop instruments, such as conventional potentiostats, have improved with micro- and nanofabrication to create electroanalytical tools as small as a cellphone, allowing people to perform experiments in the field without special training.(Ahmed et al. 2016) One of the best examples of a POC electrochemical biosensor is the glucometer, introduced by Clark et al. in 1962.(Clark Jr. and Lyons 1962) These widely used diagnostic devices are coupled to a pocket-sized amperometric transducer,(Wang 2006) and have been the leading diabetes monitoring devices in the market over the past decades, used both by physicians and patients.(Lan, Zhang, and Lu 2016) Additionally, electrochemical biosensors possess most of the ASSURED criteria set by the WHO to qualify as efficient diagnostic testing: Affordable, Sensitive, Specific, User-friendly, Rapid and robust, Equipment-free and Deliverable to end-users, making them prime candidates for SARS-CoV-2 detection.(Pokhrel, Hu, and Mao 2020)

We hope you will find our revised manuscript suitable for publication in *Nature Communications*.

Response to comments of Reviewer #1

REVIEWER COMMENTS

Thank you for resubmitting the manuscript and readjustment of error bar for figure 2 and 3. addressing the comments. However, I have identified a-few discrepancies in various version of the submitted manuscript that needs detailed clarification. Please see below.

Figure 5a: looking at the data submitted in three different time the value of histogram keeps changing but the number of negative patient data in table 5c showed that the same number of negative patients were tested each time. What are the criteria for eliminating the data and readjusting the values and error bars.

Figure 5b: number of data point keep changing from first to last submission of the data. Is there any specific criteria or methodology used for data removal or entry of new data. Why data was removed?

Figure 5d: the value for first submission has error bars but second submission it was removed. In third submission the mean of highest value of peak shift changed (earlier two submission it was approximately 60 while in recent submission the value is less than 60). What are the criteria for data readjustment in each submission.

In Figure 4d-f: the peak shift values and error bar for figure 4d to 4f keep changing from first to second to recent submissions. I can see that trend of peak shift for Figure 4d changed from first to second submission of figure. Did author repeated the experiment and new data submitted? In figure 4f the error bar also keeps changing with each submission.

Figure 5f: the values of different histogram keep changing with each submission. How the values were readjusted for each submission.

Figure 5g: it was removed from first submission to second submission of the data. Removal and addition of any data need to be clarified.

Response:

Over the course of the project we've handled data analysis by several alternative methods:

1. Showing anodic peak shifts.
2. Showing cathodic peak shifts.
3. Showing average peak shifts from first and second CV sweeps.

As the project went on, we found that the cathodic peak shift is the most reliable analysis method, and chose it as the defining signal, however when going back through the data, after the editorial request on data analysis clarifications and raw data explanations, several plot points were missed

in the re-calculation for cathodic peaks only, in several instances still showing the anodic shift or average value instead. Thus, we corrected these cases for consistency. Of course, the raw data remains the same in all instances in this case.

Furthermore, when we first started compiling the raw data, and while going over the data a second and third time, after scrutinizing the data during the revision rounds requested by reviewers and in particular the last rounds of editorial requests for data analysis and raw data analysis explanations, we found several errors that needed to be fixed, caused during the compilation of large data volume in several graphs, mainly caused by copy/paste errors. All these errors do not change at all the interpretation of the data, and the corrections were applied when errors were found during the data repeated analysis and verification along the multiple rounds of revisions.

We want to emphasize that the last raw data files provided are now completely accurate and represent the correct revised data.

Also, all changes made by us while scrutinizing the data along the several revisions rounds were a result of i.e. the addition of further replicates, changes to data analysis approaches to ensure consistency, and correcting identified errors in the data processing. We explained these changes in numerical detail to the editorial team, and are now confident that all data presented in the manuscript, and the corresponding source data are correct.

We feel confident that the manuscript is now greatly improved thanks to the reviewers and excellent editorial work, which helped us in verifying that all the presented data is completely accurate along the many rounds of revision, and believe you will find our response clear enough in order to move forward with final acceptance of the manuscript.